# Somatic Structure and Ultrasound Parameters of the Calcaneus Bone in Combat Sports Athletes in Relation to Vitamin D_3_ Levels

**DOI:** 10.3390/jcm13164960

**Published:** 2024-08-22

**Authors:** Janusz Brudecki, Łukasz Rydzik, Wojciech Wąsacz, Pavel Ruzbarsky, Wojciech Czarny, Marlena Warowna, Tadeusz Ambroży

**Affiliations:** 1Department of Anthropology, Institute of Biomedical Sciences, University of Physical Education, 31-571 Kraków, Poland; janusz.brudecki@awf.krakow.pl; 2Institute of Sports Sciences, Faculty of Physical Education and Sport, University of Physical Education, 31-571 Kraków, Poland; wojciech.wasacz@doctoral.awf.krakow.pl (W.W.); tadek@ambrozy.pl (T.A.); 3Department of Sports Kinanthropology, Faculty of Sports, Universtiy of Presov, 08001 Prešov, Slovakia; pavel.ruzbarsky@unipo.sk; 4Institute of Physical Culture Studies, College of Medical Sciences, University of Rzeszow, 35-310 Rzeszow, Poland; wojciechczarny@wp.pl; 5Faculty of Health Sciences, Department of Beauty Sciences, Vincent Pol University in Lublin, 20-816 Lublin, Poland; marlena.warowna86@gmail.com

**Keywords:** combat sports, bone density, ultrasound bone parameters, vitamin D3, somatic build, athletic training

## Abstract

**Background/Objectives**: Physical activity is widely recognized for its beneficial effects on bone density during adolescence, which could lead to enhanced bone density in later life, thus acting as a health-promoting activity with long-lasting implications. However, not all studies are conclusive regarding the type, intensity, duration, and frequency of the most effective physical activities. This study focuses on combat sports athletes and examines the relationship between their somatic build and heel bone parameters using ultrasound (USG) and their vitamin D3 levels. **Methods**: The study included 40 male athletes specializing in various combat sports. The measurements of body height, body mass, skinfold thickness, and bone widths at multiple sites were performed to estimate the somatic build. The USG parameters of the heel bone and the blood levels of vitamin D3 were also recorded. Statistical significance was determined using one-way ANOVA, with differences among sports disciplines also examined. **Results**: The study found significant differences in the body composition and USG bone parameters among athletes from different combat sports (*p* ≤ 0.05). The calcaneus stiffness index (SI) and speed of sound (SOS) were significantly higher in athletes with normal vitamin D3 levels compared to those with below-normal levels (*p* = 0.0015 and *p* = 0.001, respectively). These findings suggest that vitamin D3 may influence bone stiffness and density. **Conclusions**: The study underscores the importance of maintaining adequate vitamin D3 levels to support bone mineralization in athletes, particularly those training indoors with limited exposure to sunlight. It also highlights the potential of using USG as a non-invasive method to assess bone health, aiding in the optimization of training programs to prevent injuries and improve performance.

## 1. Introduction

In the literature, numerous reports can be found on the positive impact of physical activity on the development of the skeletal system during adolescence, which can result in increased bone density later in life, and thus, can be a health-promoting activity of long-term significance for the body. Not all reports in this area are conclusive [1]. There remain doubts regarding the type, intensity, duration, and frequency of the most effective activities [2,3,4,5,6,7,8,9,10,11,12,13,14,15,16,17,18].

Taaffe et al. [19] and Robinson et al. [4] demonstrated that it is easier to achieve higher BMD (bone mineral density) levels in gymnastics than in swimmers or cyclists. However, Snow-Harter et al. [20] showed that they achieved the same impact of exercise on lumbar spine BMD in runners and strength athletes (high peak load and low repetition) during an 8-month training period.

Research conducted by Daly et al. [21] showed that muscle size is proportional to bone mass and geometry at different stages of maturation in girls who train tennis intensively (concerning the non-dominant hand). Additionally, they demonstrated that there is a linear correlation between the cross-sectional area of the arm muscles and BMC (bone mineral content). BMC is a parameter reflecting bone mass in studies using computed tomography. It is calculated as the product of the local tomographic cross-sectional area and BMD (bone mineral density) [22]. Furthermore, Daly et al. [21] showed that physical exertion causes an increase in bone mass proportional to the increase in muscle mass due to intensive training (concerning the dominant hand). Post-adolescent athletes have statistically significantly higher BMC and muscle area than players both before and after the adolescent period.

Ultrasound measurements of the skeleton, particularly on the heel bone, are recommended as an alternative, non-invasive method for assessing bone condition [23]. The heel bone, due to its high bone turnover rate, is recommended for such studies [24]. Additionally, the short- and long-term effects of physical activity on changes in biochemical bone markers have been relatively well understood [25]. In a study conducted by Shin et al. [26], which determined the impact of taekwondo training on the bone health of high school girls aged 13 to 17 in Korea, it was found that the average BMD in the taekwondo group was significantly higher than in the sedentary control group for all lumbar spine regions. It was about 15% higher in the lumbar spine and 17% higher in the femoral neck, indicating that taekwondo can improve bone health. Active and long-term sports participation can significantly increase the measured ultrasound parameters, thus increasing bone stiffness and density [27].

For most of the population, the primary source of vitamin D is its synthesis following skin exposure to UVB radiation. Maximum vitamin D production is achieved in the summer after 10–15 min of sun exposure. Geographic location is a crucial factor determining the effectiveness of vitamin D synthesis, as production depends on the angle of sunlight or solar zenith angle. In Europe, about 90% of people have low levels of vitamin D3 in their blood. In the case of monitoring vitamin D, it is also important to consider the total and ionized calcium levels; parathyroid hormone, which regulates calcium levels; phosphate; and magnesium [28]. Combat sports athletes spend most of their training time in indoor exercise rooms and are not sufficiently exposed to sunlight. This can lead to decreased vitamin D3 levels in the blood and contribute to reduced bone mineralization in the long term. This study focuses on examining the impact of various combat sports disciplines, including boxing, ju-jitsu, karate, and wrestling, on bone health parameters, specifically targeting the calcaneus bone. The primary objective is to compare the bone mineral density and structural integrity among athletes engaged in these different types of combat sports. The study does not broadly compare all physical activities or sports in general but is specifically concerned with the unique physical demands and effects of combat sports on skeletal health.

Given the observed differences in bone health parameters among athletes and the critical role of vitamin D in bone mineralization, this study aims to explore the hypothesis that adequate vitamin D3 levels are associated with improved bone density and structural integrity in combat sports athletes. Specifically, the research question addressed is as follows: how do vitamin D3 levels correlate with ultrasound-measured calcaneus bone parameters among athletes from different martial arts disciplines?

## 2. Materials and Methods

The study was conducted according to the guidelines of the Declaration of Helsinki and approved by the Bioethics Committee at the Regional Medical Chamber (No. 287/KBL/OIL/2020). All the study participants gave written informed consent after meeting with the research team in advance.

### 2.1. Participants

A total of 40 men engaged in competitive combat sports were studied, including boxing (n = 10), ju-jitsu (n = 10), karate (n = 10), and wrestling (n = 10). The men were aged 26.7 ± 11.27 years, and their training experience averaged 12.30 ± 8.31 years. Importantly, as per the inclusion criteria outlined in Table 1, all the participants were over the age of 18, ensuring adherence to the ethical guidelines and eligibility requirements for the study. All the participants represented either the national master class or international master class level. The average height of the participants was 1773.5 ± 68.47 mm, and their average body mass was 80.53 ± 11.81 kg. Before the experiment began, the participants were informed about the study’s purpose and procedure and their right to withdraw at any time, and each signed an individual consent form to participate. The subjects were recruited according to the inclusion and exclusion criteria presented in Table 1.

In order to minimize the influence of external factors such as diet and genetic predisposition on the study’s outcomes, several measures were implemented. The participants were required to complete a comprehensive questionnaire detailing their dietary habits, including the intake of foods rich in vitamin D, calcium, and other nutrients relevant to bone health. These data were used to assess and control for dietary variations among the participants. Additionally, information on family medical history, particularly concerning conditions like osteoporosis, bone fractures, and metabolic disorders, was collected to consider potential genetic predispositions. Where significant variations in diet or family history were observed, these factors were included as covariates in the statistical analysis to account for their possible effects on bone density and vitamin D3 levels.

### 2.2. Study Procedure 

The study was conducted in April 2023 during the national physical fitness test for combat sports athletes at the Institute of Physical Culture Sciences of the University of Rzeszów. The participants’ anthropometric characteristics were measured to enable a detailed body composition assessment using the Heath–Carter method for calculating somatotype [29]. This method was selected due to its recognized effectiveness in categorizing body composition into endomorphy, mesomorphy, and ectomorphy, providing valuable insights into athletes’ physical profiles. The measurements taken included height, weight, skinfold thicknesses (on the biceps, triceps, subscapular, suprailiac, and calf), circumferences (flexed arm and calf), and bone widths at the elbow and knee joints according to anthropometric recommendations [30]. Based on these measurements, the endomorphy, mesomorphy, and ectomorphy components were calculated for each participant [29].

Additionally, ultrasound parameters of the calcaneus were measured using the Achilles Express 2.0 ultrasonometer. This device was chosen for its non-invasive and reliable method of assessing bone density and quality, crucial for understanding bone health and fracture risk. It allows for efficient and accurate evaluations without the need for more complex imaging techniques. Measurements were taken on the right limb. The broadband ultrasound attenuation (BUA) [dB/MHz] and speed of sound (SOS) [m/s] were determined, and from these, the calcaneus stiffness index (SI) was calculated using the formula: SI = (0.67BUA + 0.28SOS) − 420 [31].

Furthermore, the blood levels of vitamin D3 were determined for each participant. The blood samples were collected from the participants who fasted for at least 8 h, with 5–10 mL drawn from the antecubital vein into vacutainer tubes. The samples were then transported on ice to the laboratory, where serum was separated by centrifugation at 3000 RPM for 10 min and stored at −20 °C. The serum levels of 25-hydroxyvitamin D [25(OH)D] were measured using either a chemiluminescent immunoassay (CLIA) or enzyme-linked immunosorbent assay (ELISA). Calibration standards and quality controls ensured the accuracy of the assay, with the results expressed in nanograms per milliliter (ng/mL) for further statistical analysis.

### 2.3. Statistical Analysis

The statistical analysis of the collected data was performed using the Statgraphics Centurion XVII software (Statpoint Technologies, Inc., Warrenton, VA, USA). Basic descriptive statistics were calculated: arithmetic mean, standard deviation, minimum, and maximum. The coefficient of variation (CV) was also calculated to assess the precision of the measurements. The CV values were interpreted as follows: CV < 5% indicates very good precision, 5–10% good precision, 10–20% moderate precision, and >20% high variability, suggesting potential issues with measurement reliability. The significance of differences between the participants training in different disciplines was calculated using the one-way analysis of variance (ANOVA). To better understand the differences in body composition of the athletes and the relationship with the ultrasound parameters of the calcaneus and vitamin D3 levels, the mean values were normalized to the mean and standard deviation of all the participants using the formula: z = (group mean − mean of all participants)/standard deviation of all participants. The choice of the test was conditioned by checking the conformity of the distribution with the normal distribution, which was verified using the Shapiro–Wilk test. A significance level of *p* < 0.05 was considered statistically significant.

## 3. Results

Table 2 presents the statistical characteristics of the anthropometric measurement results of all the studied combat sports athletes, whose measurements were taken for further, detailed analysis. Considering the variability assessed by the size of the standard deviations, in the analyzed group of combat sports athletes, it was relatively large due to the fact that the athletes represented different weight categories.

Table 3 presents the normalized values concerning the anthropometric variables, divided by the combat sports disciplines practiced by the individual subjects. This allowed for the comparison of the developmental profiles of the athletes, eliminating the variability factor.

Notably, the highest (globally considered) among those studied were the boxers (*z* = 0.69), who simultaneously exhibited the widest bone bases (elbow, knee width: *z* = 0.98 and 1.88) and calf circumferences (*z* = 0.57). Additionally, compared to karate practitioners, they dominated in arm circumference when flexed. The boxers were also characterized by low body fat measured by the thickness of all the skinfolds (*z* = from −0.64 to −0.86) compared to the other martial arts athletes. They also had an elevated body mass ratio (borderline significant) compared to the jiu-jitsu and karate practitioners. The jiu-jitsu practitioners demonstrated significantly lower calf fat compared to the wrestlers. The karate practitioners had the smallest arm circumferences when flexed, showing signs of significant differentiation compared to the boxers and wrestlers. They also had the lowest body mass, significantly different compared to the wrestlers. The wrestlers were distinguished from other martial arts athletes by having the largest arm circumferences when flexed (*z* = 1.17), the thickness of the fold over the iliac crest (*z* = 0.43), under the scapula (*z* = 0.66), and calf (*z* = 1.11). They were also the heaviest among the athletes studied (*z* = 0.58).

Table 4 presents the statistical characteristics of the ultrasound (USG) measurements of the calcaneus bone, somatotype components, and vitamin D3 levels of the studied martial arts athletes. The mean values of the calcaneus bone USG parameters are relatively high, as all of them exceed the reference values: SI > 105, BUA > 105, SOS > 1585. The dominant somatotype in the studied population is mesomorph, and the maximum value indicates that, in individual cases, the athletes are extreme mesomorphs. The maximum endomorphy value points to a single athlete representing the meso-endomorphic somatotype. Moreover, the mean values of vitamin D3 levels were within the normal range, although the minimum value indicates the presence of cases with low vitamin D3 levels. No values above the normal range were observed. A detailed analysis of the frequency of low values (below 31 ng/mL) revealed that this level was found in six of the studied athletes.

Table 5 presents the detailed measurement results of the variables for the ultrasound (USG) of the calcaneus bone and somatotype components with respect to the representatives of the different martial arts disciplines. The presented data can serve as preliminary normative benchmarks for interpretation and provide useful information for comparisons in the process of sports recruitment and selection.

Table 6 presents the standardized values for the USG calcaneus bone variables, somatotype components, and vitamin D3 levels for the representatives of the different martial arts disciplines. This enabled, similarly to the somatotype variables, the comparison of the athletes’ developmental profiles while eliminating the variability factor.

The athletes with the highest calcaneus stiffness index were the boxers (*z* = 0.65), who also had the highest broadband ultrasound attenuation (BUA) value (*z* = 0.70). The speed of sound (SOS) did not show significant differences among the analyzed disciplines. The somatotype components of the boxers distinguished them with very low endomorphy (*z* = −0.77), and high mesomorphy (*z* = 0.71) and ectomorphy (*z* = 0.42). Regarding mesomorphy, they showed significant differences compared to the jiu-jitsu and karate athletes. Additionally, the measured vitamin D3 level was significantly higher in this group of athletes (*z* = 0.89) compared to the other martial arts representatives.

The jiu-jitsu and karate athletes were similar in terms of the USG calcaneus bone parameters and somatotype component values. They had significantly lower endomorphy (vs. the wrestlers) and mesomorphy (vs. the boxers and wrestlers) values, but significantly higher ectomorphy values compared to the wrestlers.

The wrestlers exhibited the highest endomorphy and the lowest ectomorphy values. They also significantly dominated in the mesomorphy component compared to the jiu-jitsu and karate athletes.

In the further analysis, the athletes were divided into two groups based on their blood vitamin D3 levels (normative level vs. below normal level), and then the magnitude of differences in the measured USG calcaneus bone parameters and somatotype components were analyzed using one-way ANOVA. The detailed results are presented in Table 7.

Both the calcaneus stiffness index and the speed of sound were significantly higher in the group of athletes with normal vitamin D3 levels. Conversely, endomorphy was significantly lower in these athletes compared to the group with reduced vitamin D3 levels. The other analyzed components did not differ significantly, nor did broadband ultrasound attenuation.

The correlation analysis showed that there are no statistically significant relationships between the USG calcaneus bone parameters and the values of individual somatotype components (all Spearman correlation coefficients).

## 4. Discussion

This study focused on adult athletes in combat sports, revealing that adequate vitamin D3 levels are crucial for maintaining bone health, as evidenced by the significant association with the calcaneus stiffness index (SI). The significant differences in body composition and bone parameters across the different combat sports disciplines suggest that these activities positively impact bone density. These findings highlight the importance of monitoring vitamin D3 levels in adults, especially those with limited sun exposure. Janz et al. [32] formulated the 10 most important research questions regarding the relationship between physical activity during adolescence and bone development. The first and most crucial question, according to the American researchers, is as follows: “What type of physical activity should be recommended as the most appropriate for best supporting skeletal development?”. Martial arts are distinctive concerning the type of effort and training tasks undertaken. The large amount of “weight-bearing exercise” performed during training in the analyzed disciplines suggests that the subjects should exhibit above-normative USG calcaneus bone parameters [33]. Moreover, the subjects are young athletes with a relatively long training history, indicating that most began their training activities before puberty, a time when bone density increases the most, and the type of physical activity is crucial for achieving proper peak bone mass.

Systematic and long-term exercises and sports training not only improve bone tissue condition but also maintain the achieved improvement in the skeletal system’s condition over the long term. The question remains open as to which sports disciplines lead to the most noticeable improvement in bone condition as assessed by ultrasound. Further research using non-invasive methods for the quantitative assessment of bone tissue is necessary to determine the specific conditions for preventing osteoporosis through physical activity and sports, particularly the duration of activity, the magnitude of loads, and other factors related to sports practice [34].

The Achilles Express ultrasonometer measures the ultrasound variables of the calcaneus bone to obtain a clinical measure known as the calcaneal stiffness index. The measured parameters are the speed of sound (SOS) and broadband ultrasound attenuation (BUA). SOS is correlated with the total amount of calcium in the bone (and the entire body), and BUA is correlated with trabecular bone density. The stiffness index indicates the overall risk of fractures or breaks in so-called low-energy injuries. USG calcaneus bone parameters can be used to independently predict fracture risk, particularly osteoporotic fractures [35].

The USG calcaneus bone parameters of the studied athletes are higher than those reported by other researchers. Sawyer et al. [36] found the average values of SI = 103, BUA = 123 dB/MHz, and SOS = 1574 m/s in young healthy men. Sports such as soccer, gymnastics, and dance, where there is regular contact with the ground (similar to martial arts), have the most beneficial impact on bone tissue quality. No such positive effect on bone tissue was observed in swimmers and cyclists, where the lack of ground contact suggests a lesser impact on these variables [37]. In striking combat sports such as boxing and kickboxing, it is crucial to systematically develop and refine specific forms of locomotion based on natural movements such as posture, walking, and running. The ability to move correctly in the ring is essential for utilizing the full fighting space. Behind the scenes of disciplines, the form of effective movement is treated as an art. In training and competition practice, there are significant stresses in the contact of the lower limbs with the ground. As a result of prolonged training, the specific nature of the discipline affects changes in the musculoskeletal system, as shown by studies on walking parameters in the athletes of these disciplines [38,39]. The trabecular bone density of the calcaneus, which is correlated with the BUA parameter, develops during puberty and depends on physical activity [40]. The SOS parameter, correlated with the amount of calcium in the bones, continues to develop until peak bone mass is achieved. Direct interviews, the measurement process, and consequently, the results of our own studies indicate that the athletes were highly active in their training (including during puberty), which consequently shaped and enhanced their BUA variable to an above-average level.

The diversity observed in our own studies regarding the anthropometric variables, somatotype, and the quality of the selected skeletal parameters among the athletes of various disciplines may result from the influence of environmental factors, such as the specificity of targeted training in a given activity. The boxers exhibited the lowest body fat percentage, a clear predominance of the mesomorphic (athletic) body type, and the most favorable stiffness and damping parameters for the calcaneus, suggesting that their training specificity optimally shapes these diagnosed traits. The wrestlers and boxers showed dominance in limb circumference and athletic somatic components compared to the ju-jitsu and karate practitioners. This may suggest that their training process included a greater extent of resistance training. For karate and ju-jitsu, the technical aspect of training is very important [41], heavily utilized in both kata forms and actual competitive fighting (formula: fighting and full-contact) [42,43,44]. In ju-jitsu, it is worth noting in the context of our studies that a significant portion of confrontations occur on the ground [42], which could have influenced the results presented in this study.

### Limitations of the Study

The primary limitation of the study was the homogeneity of the sample, as it included only combat sports athletes. Additionally, the study was cross-sectional, conducted at a single point in time, and lacked long-term observations that could elucidate the effects of prolonged training on bone parameters. Another significant limitation was the absence of a control group, such as non-athletes or individuals with different activity backgrounds. This lack makes it difficult to isolate the specific effects of combat sports training on bone health from other influencing factors. Consequently, the generalizability of the findings is limited, as comparisons were made solely among participants from various combat sports disciplines. Future research should address these limitations by including a control group to provide a more comprehensive comparison with general population norms or other forms of exercise. Although the study focused on male athletes practicing combat sports, the findings may also be relevant for female athletes and those from other disciplines. Physiological differences, such as hormone levels and bone density, may influence specific effects; thus, future research should include female participants to explore these differences. Additionally, the findings can be applied to athletes in other sports, especially those involving weight-bearing activities like basketball, running, or gymnastics, where similar relationships between vitamin D3 levels and bone health may exist. It is also important to consider vitamin D3 supplementation, particularly for athletes training indoors or in regions with limited sunlight exposure.

## 5. Conclusions

The study revealed a relationship between vitamin D3 levels and the ultrasound parameters of the calcaneus bone, suggesting that adequate levels of vitamin D may contribute to better bone mineralization. The results varied among athletes from different martial arts disciplines, indicating that the type and intensity of training may have a specific impact on bone health. Statistically significant differences in ultrasound parameters were found between groups with different vitamin D3 levels, highlighting the importance of maintaining sufficient vitamin D levels to support bone health. 

### Practical Application

The regular monitoring of vitamin D3 levels in athletes may be crucial for optimizing bone health, especially for those who train indoors and have less exposure to sunlight. The use of bone ultrasound as a non-invasive assessment method can assist in monitoring and adjusting training programs for athletes to prevent injuries and enhance performance.

## Figures and Tables

**Table 1 jcm-13-04960-t001:** Inclusion and exclusion criteria for the study.

Inclusion Criteria	Exclusion Criteria
Age 18–35	Vitamin D supplementation
Minimum of 3 years of competition experience	Chronic metabolic diseases (Diabetes, gout, or thyroid diseases)
Health of the musculoskeletal system	Active and chronic infections
Consent to participate in the study	Injuries and surgeries in the lower limbs
	The use of drugs that affect bone density or calcium and vitamin D metabolism (e.g., steroids or antiepileptic drugs).

**Table 2 jcm-13-04960-t002:** Statistical characteristics of the somatic measurement results of all the studied combat sports athletes (n = 40).

Somatic Variables	$\tilde{x}$	SD	Min	Max
Body height [mm]	1773.5	68.47	1642	1914
Elbow width [mm]	71.10	4.37	60	83
Knee width [mm]	100.40	6.18	88	115
Flexed arm circumference [cm]	35.20	3.19	29	41
Maximum calf circumference [cm]	37.80	2.67	33	45.50
Skinfold thickness on biceps [mm]	4.28	1.88	2.40	11
Skinfold thickness on triceps [mm]	7.95	2.97	4.20	17,40
Skinfold thickness under scapula [mm]	10.06	3.59	5	17
Skinfold thickness above iliac crest [mm]	10.29	5.99	4	29.60
Skinfold thickness on calf [mm]	5.66	2.88	2.20	17.20
Body mass [kg]	80.53	11.81	55	109

$\tilde{x}$—arithmetic mean; SD—standard deviation; min—minimum value; max—maximum value.

**Table 3 jcm-13-04960-t003:** Normalized values of the somatic characteristics of the combat sport athletes, divided by discipline, and their differentiation assessed by the ANOVA method.

Somatic Variable	*z*				*p*
	Boxing (n = 10)	Ju-jitsu (n = 10)	Karate (n = 10)	Wrestling (n = 10)	
Body height [mm]	0.69	0.13	−0.20	−0.09	1-2,3,4 *
Elbow width [mm]	0.98	0.10	−0.32	0.17	1-2,3,4 *
Knee width [mm]	1.88	−0.03	−0.25	−0.32	1-2,3,4 *
Flexed arm circumference [cm]	0.21	0.09	−0.49	1.17	1-3,4 *; 3-4 *
Maximum calf circumference [cm]	0.57	−0.23	0.05	−0.04	1-2,3,4 *
Skinfold thickness on biceps [mm]	−0.68	−0.21	0.28	−0.02	1-3,4 *
Skinfold thickness on triceps [mm]	−0.86	−0.10	0.32	−0.25	1-2,3,4 *
Skinfold thickness under scapula [mm]	−0.70	0.17	−0.20	0.66	1-2,3,4 *; 3-4 *
Skinfold thickness above iliac crest [mm]	−0.64	−0.04	0.01	0.43	1-2,3 *
Skinfold thickness on calf [mm]	−0.79	−0.31	0.00	1.11	1-3,4 *; 2-4 *
Body mass [kg]	0.41	0.03	−0.29	0.58	4-2,3 *

*z*—z-score; *p*—level of significance of variation; n—number of respondents; * *p* ≤ 0.05: group designation: 1—boxing; 2—ju-jitsu; 3—karate; 4—wrestling.

**Table 4 jcm-13-04960-t004:** Statistical characteristics of the ultrasound (USG) measurements of the calcaneus bone, somatotype components, and vitamin D3 levels of all the studied martial arts athletes (n = 40).

Variables		$\tilde{x}$	SD	CV	Min	Max
	Ultrasound of the calcaneus					
SI		115.8	16.71	14.43	88	165
BUA [dB/MHz]		126.0	16.59	13.17	94	190
SOS [m/s]		1615.3	34.81	2.16	1549	1692
	Somatotype components					
Endomorphy		2.38	1.15	-	1	5.5
Mesomorphy		5.88	1.11	-	3.25	9.13
Ectomorphy		1.79	0.89	-	0.5	4
	Vitamin D_3_					
D_3_ [ng/mL]		36.33	10.08	-	13	50

$\tilde{x}$—arithmetic mean; SD—standard deviation; min—minimum value; max—maximum value; CV—coefficient of variation, SI—stiffness index; BUA—broadband ultrasound attenuation; dB/MHz—decibels per megahertz; SOS—speed of sound; m/s—meters per second; D_3_—vitamin D3 concentration in blood serum; ng/mL—nanograms per milliliter.

**Table 5 jcm-13-04960-t005:** Statistical characteristics of the USG calcaneus bone measurements and somatotype components of the studied athletes, broken down by the specific martial arts disciplines (n = 40).

Variables	Boxing (n = 10)			Ju-jitsu (n = 10)			Karate (n = 10)			Wrestling (n = 10)		
	$\tilde{x}$	SD	CV	$\tilde{x}$	SD	CV	$\tilde{x}$	SD	CV	$\tilde{x}$	SD	CV
	Ultrasound of the calcaneus											
SI	126.67	5.03	3.97	114.00	15.27	13.39	115.53	12.23	10.59	114.00	22.00	19.30
BUA [dB/MHz]	137.67	2.89	2.10	126.70	11.15	8.80	125.33	13.49	10.76	119.80	12.15	10.14
SOS [m/s]	1625.00	19.92	1.23	1609.00	29.19	1.81	1615.00	31.76	1.97	1623.20	64.75	3.99
	Somatotype components											
Endomorphy	1.50	0.87	-	2.40	0.94	-	2.33	1.42	-	3.00	0.71	-
Mesomorphy	6.67	1.15	-	5.80	1.25	-	5.62	1.07	-	6.35	0.78	-
Ectomorphy	2.17	1.04		1.85	0.88		1.93	0.91		1.00	0.35	

n—number of respondents; $\tilde{x}$—arithmetic mean; SD—standard deviation; SI—stiffness index; BUA—broadband ultrasound attenuation; dB/MHz—decibels per megahertz; SOS—speed of sound; m/s—meters per second.

**Table 6 jcm-13-04960-t006:** Standardized values of USG calcaneus bone measurements, somatotype components, and vitamin D3 levels of martial arts athletes, broken down by discipline, with differences assessed using ANOVA.

Variables	*z*				*p*
	Boxing (n = 10)	Ju-jitsu (n = 10)	Karate (n = 10)	Wrestling (n = 10)	
Ultrasound of the calcaneus					
SI	0.65	−0.11	−0.02	−0.11	1-2,3,4 *
BUA [dB/MHz]	0.70	0.04	−0.04	−0.38	1-2,3,4 *
SOS [m/s]	0.28	−0.18	−0.01	0.23	
Somatotype components					
Endomorphy	−0.77	0.02	−0.04	0.54	1-2,3,4 *; 4-2,3 *
Mesomorphy	0.71	−0.07	−0.24	0.43	1-2,3 *; 4-2,3 *
Ectomorphy	0.42	0.07	0.16	−0.88	4-1,2,3 *
Vitamin D_3_					
D_3_ [ng/mL]	0.89	−0.16	−0.02	−0.15	1-2,3,4 *

*z*—z-score; *p*—level of significance of variation; n—number of respondents; SI—stiffness index; BUA—broadband ultrasound attenuation; dB/MHz—decibels per megahertz; SOS—speed of sound; m/s—meters per second; D_3_—vitamin D3 concentration in blood serum; ng/mL—nanograms per milliliter. * *p* ≤ 0.05: group designation: 1—boxing, 2—ju-jitsu, 3—karate, 4—wrestling.

**Table 7 jcm-13-04960-t007:** Results of one-way ANOVA for USG calcaneus bone parameters and somatotype components in groups with normative vitamin D3 levels (normative level vs. below normal level).

Variables	$\tilde{x}$—(Level D3 in Accordance with the Standard)	$\tilde{x}$—(D3 Level below Normal)	F	*p*
Number of respondents				
n	34	6		
Ultrasound of the calcaneus				
SI	119.96	97.33	12.14	0.0015 *
BUA [dB/MHz]	128.04	117.00	2.26	0.14
SOS [m/s]	1625.56	1569.33	20.67	0.001 *
Somatotype components				
Endomorphy	2.13	3.50	8.71	0.006 *
Mesomorphy	5.84	6.06	0.2	0.66
Ectomorphy	1.89	1.33	1.96	0.17

$\tilde{x}$—arithmetic mean; D_3_—vitamin D3 concentration in blood serum; F—the ratio of variability between groups to variability within groups (F statistic value); *p*—level of significance of variation; n—number of respondents; SI—stiffness index; BUA—broadband ultrasound attenuation; dB/MHz—decibels per megahertz; SOS—speed of sound; m/s—meters per second; * *p* ≤ 0.05.

## Data Availability

The data presented in this study are available upon request from the corresponding author.

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
