# Peer review of "Somatic Structure and Ultrasound Parameters of the Calcaneus Bone in Combat Sports Athletes in Relation to Vitamin D3 Levels"

_jcm, 2024, doi:10.3390/jcm13164960_

Round 1

Reviewer 1 Report

Comments and Suggestions for Authors

- Remove substitution with vitamin D from the inclusion criteria since it has already been stated that it is an exclusion criterion!

- Name the chronic metabolic diseases of interest!

- Specify parameters that are of interest to the value of vitamin D, such as Ca, phosphate!

- Report the conclusions as a text unit, without separating them!

Author Response

Dear Reviewer,

Thank you very much for your time and valuable comments, which all have been considered and incorporated. The detailed list of responses is given below. We hope that the modifications and explanation will be acceptable for you.

Yours sincerely,

Rydzik, corresponding author

  • Remove substitution with vitamin D from the inclusion criteria since it has already been stated that it is an exclusion criterion!
  • A: This has been corrected 
  • Name the chronic metabolic diseases of interest!
  • A: This has been corrected 
  • Specify parameters that are of interest to the value of vitamin D, such as Ca, phosphate!
  • A: This has been corrected 
  • Report the conclusions as a text unit, without separating them!
  • A: This has been corrected 

Reviewer 2 Report

Comments and Suggestions for Authors

a. there is a lack of hypothesis or research question at the end of the introduction

b. in the introduction, specify whether the study is comparing different sports, physical activities in general, or a type of activity

c. acknowledge the lack of a control group in the limitations section

d. describe the procedure for collecting and analyzing blood samples for vitamin D3 levels

e. please specify the reason for Heath-Carter method for somatotype and Achilles Express 2.0 ultrasonometer for calcaneus measurements

f. how did the authors ensure that the participants fully understood the study and provided informed consent?

g. can the authors detail on how potential factors (diet and genetic predisposition) were controlled in their analysis?

h. how can the study findings be applied to female athletes or to athletes from other sports disciplines?

Author Response

Dear Reviewer,

Thank you very much for your time and valuable comments, which all have been considered and incorporated. The detailed list of responses is given below. We hope that the modifications and explanation will be acceptable for you.

Yours sincerely,

  1. there is a lack of hypothesis or research question at the end of the introduction

A: This has been corrected 

  1. in the introduction, specify whether the study is comparing different sports, physical activities in general, or a type of activity

A: This has been corrected 

  1. acknowledge the lack of a control group in the limitations section

A: This has been corrected 

  1. describe the procedure for collecting and analyzing blood samples for vitamin D3 levels

A: This has been corrected 

  1. please specify the reason forHeath-Carter method for somatotype and Achilles Express 2.0 ultrasonometer for calcaneus measurements

A: This has been corrected 

  1. how didthe authorsensure that the participants fully understood the study and provided informed consent?

A: This has been corrected 

  1. can the authors detail on how potential factors (diet and genetic predisposition) were controlled in their analysis?

A: This has been corrected 

  1. howcanthe study findings be applied to female athletes or to athletes from other sports disciplines?

A: This has been corrected 

Reviewer 3 Report

Comments and Suggestions for Authors

Thanks for this interesting study.

It is exciting and important to investigate the relationship between bone density and vitamin D levels, especially in athletes who work indoors and engage in various combat sports.

I suggest several corrections.

It would be more accurate to share numerical data while reporting the results of the study in the Results subheading in the Abstract section—p values, correlation coefficients, etc.

Is there an unnecessary use of references in the first paragraph of the introduction? [2-18] is a pervasive use of references. Are all of these 17 articles related to this paragraph? I think your use of more specific references would be more accurate.

Paragraphs 2 and 3 of the introduction contain only literature information. It may be more accurate to be found in the discussion section. 

The purpose is clearly stated, and the study question is clearly stated. Thank you.

Your study group was 26.7 ± 11.27 years old. Does this mean that patients younger than 18 years of age were included in the study? In the inclusion criteria you mentioned in Table 1, the lower age limit is stated as 18. Please also state this in the text, as this is an important parameter. 

"Based on these measurements, the endomorphy, mesomorphy, and ecto-109 morphy components were calculated for each participant." Please provide reference

“SI=(0.67BUA+0.28SOS)-420”. Please provide a reference,

According to the rules of English grammar, it is recommended that each paragraph should consist of at least 3 sentences. Please combine the single-sentence paragraphs in the Results section appropriately.

Your tables in Results are very well organized and constructed. Congratulations.

It would be more accurate to mention the most important point of your study and your most critical findings in the first paragraph of the Discussion section. In the first paragraph of your discussion, the study question "What type of physical activity should be recommended as the most appropriate for best supporting skeletal development in children and adolescents" is emphasized. However, all of the patients in your study are over 18 years of age. This creates confusion. You should emphasize more accurately in the first paragraph.

The length and content of the discussion is strong and sufficient

The study question was answered correctly and sufficiently.

The organization of the study is successful

Thanks to the authors

Comments on the Quality of English Language

According to the rules of English grammar, it is recommended that each paragraph should consist of at least 3 sentences. Please combine the single-sentence paragraphs in the Results section appropriately.

Author Response

Thanks for this interesting study.

It is exciting and important to investigate the relationship between bone density and vitamin D levels, especially in athletes who work indoors and engage in various combat sports.

I suggest several corrections.

It would be more accurate to share numerical data while reporting the results of the study in the Results subheading in the Abstract section—p values, correlation coefficients, etc.

A: This has been corrected

Is there an unnecessary use of references in the first paragraph of the introduction? [2-18] is a pervasive use of references. Are all of these 17 articles related to this paragraph? I think your use of more specific references would be more accurate.

A: We suggest leaving as this is a literature review , Once removed, the introduction would be much shorter 

Paragraphs 2 and 3 of the introduction contain only literature information. It may be more accurate to be found in the discussion section. 

A: The discussion has been improved 

The purpose is clearly stated, and the study question is clearly stated. Thank you.

A: Thank you

Your study group was 26.7 ± 11.27 years old. Does this mean that patients younger than 18 years of age were included in the study? In the inclusion criteria you mentioned in Table 1, the lower age limit is stated as 18. Please also state this in the text, as this is an important parameter. 

A: This has been corrected

"Based on these measurements, the endomorphy, mesomorphy, and ecto-109 morphy components were calculated for each participant." Please provide reference

A: This has been corrected

“SI=(0.67BUA+0.28SOS)-420”. Please provide a reference,

A: This has been corrected

According to the rules of English grammar, it is recommended that each paragraph should consist of at least 3 sentences. Please combine the single-sentence paragraphs in the Results section appropriately.

A: This has been corrected

Your tables in Results are very well organized and constructed. Congratulations.

A: Thank you 

It would be more accurate to mention the most important point of your study and your most critical findings in the first paragraph of the Discussion section. In the first paragraph of your discussion, the study question "What type of physical activity should be recommended as the most appropriate for best supporting skeletal development in children and adolescents" is emphasized. However, all of the patients in your study are over 18 years of age. This creates confusion. You should emphasize more accurately in the first paragraph.

A: This has been corrected

The length and content of the discussion is strong and sufficient

A: Thank you 

The study question was answered correctly and sufficiently.

A: Thank you 

The organization of the study is successful

A: Thank you 

Thanks to the authors

A: Thank you 

Comments on the Quality of English Language

According to the rules of English grammar, it is recommended that each paragraph should consist of at least 3 sentences. Please combine the single-sentence paragraphs in the Results section appropriately.

A: This has been corretced